# Can ChatGPT Help in Electronics Research and Development? A Case Study with Applied Sensors

**DOI:** 10.3390/s23104879

**Published:** 2023-05-18

**Authors:** Zoltán Tafferner, Balázs Illés, Olivér Krammer, Attila Géczy

**Affiliations:** 1Department of Electronics Technology, Faculty of Electrical Engineering and Informatics, Budapest University of Technology and Economics, H-1111 Budapest, Hungaryilles.balazs@vik.bme.hu (B.I.); krammer.oliver@vik.bme.hu (O.K.); 2LTCC Technology Research Group, Łukasiewicz Research Network—Institute of Microelectronics and Photonics, 02-255 Kraków, Poland

**Keywords:** generative large language model, ChatGPT, sensors, electronics, engineering, development, IoT

## Abstract

In this paper, we investigated the applicability of ChatGPT AI in electronics research and development via a case study of applied sensors in embedded electronic systems, a topic that is rarely mentioned in the recent literature, thus providing new insight for professionals and academics. The initial electronics-development tasks of a smart home project were prompted to the ChatGPT system to find out its capabilities and limitations. We wanted to obtain detailed information on the central processing controller units and the actual sensors usable for the specific project, their specifications and recommendations on the hardware and software design flow additionally. Furthermore, an extensive literature survey was requested to see if the bot could offer scientific papers covering the given topic. It was found that the ChatGPT responded with proper recommendations on controllers. However, the suggested sensor units, the hardware and software design were only partially acceptable, with occasional errors in specifications and generated code. The results of the literature survey showed that non-acceptable, fabricated citations (fake authors list, title, journal details and DOI—Digital Object identifier) were presented by the bot. The paper provides a detailed qualitative analysis, a performance analysis and critical discussion of the aforementioned aspects while providing the query set, the generated answers and codes as supplied data with the goal to give added value to electronics researchers and developers if trying to reach out for the tools in their profession.

## 1. Introduction

ChatGPT is a variation of the large language model (LLM) named GPT-3 from OpenAI, trained on a given dataset [1]. ChatGPT, which is based on GPT-3 using a specific algorithm for learning, could be used for brainstorming in a conversational way [2], and as it turns out from recent literature, its possibilities can bring a revolution to working environments in various professions. However, considering the recent hype around it can also mean false recognition of its current capabilities. While the application of ChatGPT seems to spread in a wide spectrum, there are also concerns that were already raised in the scientific society.

Most of the recent papers (as seen below) present a serious concern: While generative chatbots are transformative tools of the future, urgent guidelines are needed in the case of most professions to mitigate the problems arising with the uncertainties around them [1]. It is also a serious problem that sometimes it gives convincing but false information (reported by many of the papers) and that the tool could not provide in-depth analysis. Further concern centers on the possible future monetization strategies around the tool, which could widen the gap between researchers of different wealth. It could also raise issues of cheating and unoriginal content in education and scholarly assignments. However, the tool could accelerate research, improve workflow, lower language barriers or show new ideas.

### 1.1. Aspects of LLM in Scientific Works

One significant issue is that ChatGPT might pose a danger from a scientific publishing point of view as it can rewrite content to make plagiarism detection nearly impossible. There are tools to mitigate this problem, to detect AI generated text. Despite the fact that these applications are being developed to help detect the unethical use of generative AI, they are not functioning yet perfectly, and a simple rewording by the user could fool the detection systems [3]. It was highlighted by Bhattacharya et al. that ChatGPT works from existing text, and it can produce output that comes out negatively from a plagiarism tracker [4]. Aljanabi et al. reported that the output from ChatGPT is not always accurate, but the team emphasized that it can be helpful in academic writing (getting references, rewriting or generating text, proposing style, etc.). ChatGPT can function as a search engine [5]. It was noted that it cannot handle mathematical calculations and other types of specific queries. The widely cited help that ChatGPT can provide needs further assessment in our opinion.

According to the authors, the tool could be useful for detecting security vulnerabilities as well. While the AI is supposed to be able to generate a literature review (or even complete papers and arguments [6]), it actually lacks an understanding of the implicit ideas in a text [7], which are incredibly important in human–human communication. This phenomenon could introduce errors to the answer. Also, intensive “Socratic” questioning can make ChatGPT change its answers, suggesting that they were not logically coherent or lacked sufficient proof and founding [8]. The tool often provides convincing text that can contain false parts in its details [4], and due to the black-box setup of the AI tools [9], transparent mitigation is difficult. Another issue of using an LLM, like ChatGPT, is that it is trained on a very large dataset. While this data is supposed to be diverse, there is no guarantee of a diversity of opinions. Therefore, the dataset should include biases, those could propagate through the AI and appear in the text generated [10], without the knowledge of the users. Bias, commercialization and technical precision should be investigated for a wide array of scientific and technical fields and professions.

### 1.2. Application in Medical Sciences

From a medical application point of view, ChatGPT could be useful for treating patients in surgery (analyzing vital signs, pain tolerance, medical history, facilitating communication, etc.). It could help doctors make a diagnosis without having to do an extensive manual literature review [11,12]. Macdonald et al. used the tool for a socio-medical experiment [13]. They simulated virus spreading among a population, and ChatGPT was asked to help with determining the effectiveness of the fictive vaccine and drafting a research paper. ChatGPT generated abstracts passed plagiarism. After the dataset being described, ChatGPT could explain and offer potential study steps, and ChatGPT could generate code for studying the dataset. It contained some faults, but after feedback, it could self-correct itself. As for manuscript writing, ChatGPT generated a coherent abstract for the paper. It was possible to use ChatGPT to do a literature review, but in this example, it provided faulty information. This important aspect needs to be assessed in other fields as well.

Macdonald found that ChatGPT could become a resource for researchers to develop papers and studies faster but with careful assessment of the given answer. Khan et al. found that ChatGPT is useful for checking grammar in written text, and it could be useful as a teaching tool for generating tasks in a medical environment [11]. In the reference [3], the authors also came to a similar conclusion. The tool can provide research assistance by providing text summaries, answering questions and generating bibliographies. It can translate, which could be useful for researchers who need to write in a second language and do not have a subscription tool for translation. In clinical management, it could improve documentation, support decision-making and facilitate communication with patients. There are noted deficiencies though: A lack of real understanding was reported, originating from the fixed and closed database. Besides, no data was found after 2021, and it could also generate unoriginal content with incorrect answers at times. This was also emphasized by Gordjin and Have [14]. Liebrenz showed that ChatGPT can also write an article for The Lancet Digital Health about AI and medical publishing ethics [15]. They highlighted that monetization of AI could produce a gap between researchers of different wealth. However, the availability of further monetized tools (such as ChatGPT4) was not available at the moment, so critical comparison could not have been performed by the authors.

### 1.3. Application in Finance and Education

Moving on to a different profession, Dowling and Lucey used the tool to assess their research process in finance research [16]. A literature review was done on both public (already included) and private (fed to ChatGPT) data. Idea generation, data identification, prep and testing framework proposition were done. The generated studies were stated by a board of referees to be of acceptable quality for a peer-reviewed finance journal. However, data analysis was missing from ChatGPT’s actual capabilities.

As for educational use, Rillig et al. [17] discussed the application in environmental research and education. Due to the working principle of the algorithm, biases in the training data could produce bias in the output of ChatGPT. The LLM output could easily be confused with an expert’s answer, even though it has no real understanding, so Rillig et al. also highlighted these risks during applications. ChatGPT could be used to accelerate research by outsourcing tasks to ChatGPT, improving workflow. Furthermore, it can also help non-native English speakers to write papers, develop ideas, etc. Nevertheless, ChatGPT could raise issues of cheating in education as well. The papers usually do not propose in-depth solutions for the problems; they only imply the need.

### 1.4. Application in IT and in Engineering Sciences

From the aspect of IT sciences and electrical engineering, Wang et al. noted [7] that “Stack Overflow” introduced a temporary ban on code generated by ChatGPT because of a low percentage of being totally correct. The bot gave plausible but incorrect answers during the presented discussions. Surameery and Shakor highlighted bug-fixing capabilities of the tool and suggested ChatGPT as a part of a more comprehensive debugging toolkit, not only as a sole solution used by developers [18]. Biswas showed mostly the positive aspects of the tool in programming, such as a technical query answering machine for explanations, guides and diagnosing problems [19]. Vemprala et al. showed a possible application mindset for robotics, where ChatGPT partially substitutes the engineer in the loop and where eventually a user can use the LLM as a tool to connect to further robots solving tasks [20]. The literature is very limited for electronics, software- and electrical engineering applications, which needs further investigations via various use cases and their documentations.

To see how efficiently ChatGPT can help in electronics research and development, we adapted the methodologies and approaches presented in the previous papers and investigated the applicability of ChatGPT for electrical engineering, design and development, where our focus was positioned to a widely studied topic: smart home applications with sensors. The question was: Can ChatGPT be a tool or a useful companion for an electrical engineer developer? The paper presents the experimental methodology, the query running with the bot with the supplemented data and the discussion of the results. The main novelty of our work is to show the applicability of the tool in an area that was not discussed before, and our findings might help discussions and arguments regarding the application from the professional industrial level to education and academic sciences.

## 2. Experimental Information

We wanted to investigate if ChatGPT could help us with the development steps in a smart home project. Our case study is a relevant example since smart home and IoT related developments are very popular lately (IEEE Xplore lists 13.000 articles for the term “smart home” since 2013). So, the bot could work on a well-established topic, which is also interesting from the aspect of actual hardware and software development. Furthermore, the complexity of the topic is deep enough to test the ability of ChatGPT to give answers to various layers of the problem, such as:controllers as a central unit, sensor component types, parameters, comparison of different types;logic connection between components;how can the sensors be programmed with an embedded mindset;actual code generation;and expanding the view with literature references.

We focused mainly on the electrical engineering aspects of the project, excluding data acquisition, repository, database and further IoT related questions. The workflow was the following:ChatGPT was introduced to the problem by the first query: whether to use a Raspberry Pi or an Arduino board as a sensor controller for a smart home. This question is general for a start and requires the chatbot to look up information and reason for either solution.Afterwards, more specific questions have been asked. When it was already decided which controller should be used, the bot was asked to provide a list of sensors that would be recommended for the board.Then the bot was asked to provide specific sensor models for each type of sensor, to list the sensors, to compare two alternatives or to provide the specifications and prices for the sensors.The bot was also asked about the workflow of connecting specific sensors to the controller and supporting software development.Finally, it was prompted to provide a literature review and references with DOI (Digital Object Identifier) numbers included. This step was done at the end of our queries due to the nature of the bot; the bot can dwell deeper into the topic once a human user establishes it and carries forward the discussion.

Our methodology is also presented in the following Figure 1.

ChatGPT (based on GPT-3.5, OpenAI, San Francisco, USA) was used for our experiments. The official cutoff date of the training data was in 2021 [21], which limits the current possibility of the tool. However, established information was well available before this date regarding our topic and queries. The queries were appointed to the tool between 13 and 19 March 2023, so the current work represents the state of the bot in that time. While the next version of ChatGPT Plus (GTP-4) is supposed to have better performance, it just became available during our investigations and was locked behind a paywall. Furthermore, controversies were noted, and concerns were raised regarding the new version of the tool (with an eventual ban of it in Italy) [22]. So, we aimed to stick to the more well-known version of the tool. Also, during the manuscript composition, Bard (Google) was not available in the country, meaning that general accessibility is still an issue. Bing (Microsoft) bots became available with interfacing to the internet [23], and it was freely available, so we decided to add a short (but not comprehensive) comparison between 3.5 and Bing.

## 3. Results

While it is difficult to present quantitative analysis on the problem, we try to discuss the results qualitatively, aiming for technological precision and quantitative assessment only where possible (e.g., in the case of written codes, data and specific information). The full data set (including codes, queries and responses) is available in Appendix A, which might be necessary for full assessment by the reader, as the full query text inclusion would boast the paper significantly. Appendix A availability is highlighted in the papers “Appendix A” and “Data Availability Statement” section.

### 3.1. General Questions Regarding Building a Smart Home Application, Specification of a Controller

In the first step, we asked ChatGPT which controller we should use for developing a smart home application.

***ChatGPT***: “Sure! There are several options for micro-controllers that you can choose from for your smart home project. Here are 6 examples from different manufacturers:

Arduino Uno: This is a popular micro-controller board that is easy to use and has a large community. It has 14 digital input/output pins, 6 analog inputs and a 16 MHz quartz crystal.

Raspberry Pi: This is a small, affordable computer that can run a full operating system. It has a variety of input/output options, including GPIO pins, USB ports and Ethernet.

ESP32: This is a powerful micro-controller with built-in Wi-Fi and Bluetooth connectivity. It has 34 GPIO pins, 18 analog inputs and a dual-core processor.

STM32F4 Discovery: This is a development board based on the STM32F407VGT6 micro-controller. It has 168 MHz Cortex-M4 CPU, 1 MB Flash memory, and 192 KB RAM.

PIC18F4520: This is a popular micro-controller from Microchip Technology. It has 32 KB Flash memory, 1.5 KB RAM and 36 I/O pins.

NXP LPC1768: This is a micro-controller based on the ARM Cortex-M3 architecture. It has 512 KB Flash memory, 64 KB RAM and a variety of input/output options.

These are just a few examples, and there are many other micro-controllers available on the market. You will need to choose the one that best fits your project’s requirements in terms of processing power, memory and input/output options.”

At this point, we asked ChatGPT to give arguments on choosing a device.

***ChatGPT****:* “Sure! Here are some arguments for choosing each of the micro-controllers I listed:

Arduino Uno: The Arduino Uno is a popular choice for beginners and hobbyists because it is easy to use and has a large community. There are many tutorials and examples available online, making it easy to get started. It has a simple programming environment and is compatible with a wide variety of sensors and shields.

Raspberry Pi: The Raspberry Pi is a powerful micro-controller that can run a full operating system, making it suitable for more complex projects. It has a variety of input/output options, including GPIO pins, USB ports and Ethernet, making it versatile and flexible. It can also run software like Python, making it easy to integrate with other tools and services.

ESP32: The ESP32 is a powerful micro-controller with built-in Wi-Fi and Bluetooth connectivity. This makes it ideal for projects that require wireless communication, such as remote control or monitoring. It has a large number of input/output pins and a dual-core processor, making it suitable for projects that require a lot of processing power.

STM32F4 Discovery: The STM32F4 Discovery is a development board that is based on the STM32F407VGT6 micro-controller. It has a high processing speed and a large amount of memory, making it suitable for complex projects. It also has a wide range of input/output options, including USB, Ethernet and audio.

PIC18F4520: The PIC18F4520 is a popular micro-controller from Microchip Technology. It has a relatively low cost and a large number of input/output pins, making it ideal for projects that require a lot of sensors or actuators. It also has a large community and many examples available online, making it easy to get started.

NXP LPC1768: The NXP LPC1768 is a micro-controller based on the ARM Cortex-M3 architecture. It has a large amount of memory and a variety of input/output options, making it suitable for a wide range of projects. It also has a powerful processor and can run at high speeds, making it ideal for projects that require fast processing.

In summary, each of these micro-controllers has its own strengths and weaknesses, and the best one for your project will depend on your specific requirements. Factors to consider when choosing a micro-controller include processing power, memory, input/output options and community support.”

We can say, that the information is accurate, and the bot was giving logical arguments with precise information on this very general initial case.

To narrow down the focus, we gave Arduino or Raspberry Pi as a choice. Early investigations showed that the answers give a broader point of view at the beginning of a query-set, so a narrowed approach is beneficial for a start. The bot was able to distinguish the two different systems, with reasonable responses on the general overview regarding the two systems capabilities. It gave advantages and disadvantages about the use.

***ChatGPT****:* “Both the Arduino and Raspberry Pi controllers have their advantages and disadvantages when it comes to building a smart home.

Arduino is a microcontroller board that is designed for simple projects that require low power and have limited processing requirements. It is a good choice if you want to control simple sensors such as temperature, humidity and light sensors. Arduino is easy to use and is ideal for beginners who are starting to learn about microcontrollers.

On the other hand, Raspberry Pi is a small computer that is capable of running a full operating system. It is more powerful than Arduino and has a wider range of capabilities. Raspberry Pi is a better choice if you want to integrate complex sensors such as cameras, voice recognition and machine learning. It also has the capability of connecting to the internet, which allows for remote control and monitoring of your smart home devices.

Therefore, the choice of controller depends on the complexity of your project and the sensors you want to integrate. If you are building a simple smart home system, then an Arduino controller would suffice. However, if you want to build a more advanced system that requires complex sensors and internet connectivity, then Raspberry Pi would be a better choice.”

We found that the answer contained some correct information; however, some other information can be argued with. The discussion did not go to the level of more recent advances of the systems. Arduino also offers newer devices (Portenta family [24], established in 2020) for higher-level tasks with similar capabilities to the Raspberry lineup. This information was either omitted by the timely limit of the background training data for ChatGPT 3.5 or by the fact that ChatGPT is orienting its answers towards the more common knowledge established over a longer period. According to Hernando, the origins of Arduino can be traced back to 2003–2005 [25], so the answer given by the bot was a more superficial assumption based on a longer-term general knowledge. The final conclusion of ChatGPT was found to be acceptable; most of the information was valid; and it could argue along a wide range of commercially available products without obvious bias towards one or the other.

### 3.2. Sensors and Specifications

We also asked for specific types of sensors, where the bot suggested temperature, light, motion, humidity, gas, water and smoke sensors and their combinations with short descriptions, which helps accessibility to the topic even at a beginner level. We emphasize again that, for full analysis, the reading of Appendix A is recommended.

While the scope would be too wide for analyzing all types, we focused first on temperature sensors. ChatGPT suggested five “popular” alternatives: DHT11, DHT22, LM35, DS18B20 and BME280. It provided short descriptions for each type in its answer. Each component is valid and widely used by hobbyists and professionals in the field. The query followed with light sensors. ChatGPT gave a mix of the types (e.g., photoresistors, photodiodes, phototransistors) and specific sensor modules (e.g., TSL2561, BH1740) in its five-step answer. To sum up, the bot was mixing types and specific commercial components in the answer, which is inconsistent in its manner.

Altogether, from two questions on “specific light sensors” and “specific light sensor models”, the bot listed six components and three type descriptions (with a repeated description on photoresistors). It was clear that slightly different queries affected the answer inconsistency, but the effect seemed rather minor and seemed to be originated from the randomness of each individual answer by the bot. Again, the bot was mixing types and components.

As for motion sensors, five examples were listed. The listing was presented in a third format, where the declared type was followed with a specific component name as well. (e.g., PIR Motion Sensor (HC-SR501)). However, inconsistencies were found, as for five different types, only three actual sensor components were listed, the rest was again general listing. The bot did not provide a real module for infrared motion sensors and laser motion sensors. The same mixed listing was found in the case of humidity sensors. Albeit the bot could give five actual sensor types this time, only one type was specified on its working principle (capacitive humidity sensor (SY-HS-220). Gas, water level and smoke detectors were also sorted by incoherent listing; however, all types were found to be commercially available.

ChatGPT was also asked to list the sensors by price, the bot was able to find ranges of the price, and it arranged a list based on the prices. The ranges seemed valid according to a manual search in popular web shops on the internet. All information seemed to be coherent with current availability for 10 components, but actual prices varied a bit (at the same or one range lower as the current prices). This might not only be due to the recent increase in prices—some components were inconsistently cheaper or more expensive in the list.

The bot suggested Arduino for the project previously, so the follow-up was coherently provided according to this controller type. In short, ChatGPT could carry forward the information precisely from question to question.

After the more general queries on sensors, we asked to suggest specific data for three different sensors. One query focused on the DHT11. The following specifications were given.

***ChatGPT:*** “Here are the specifications of the DHT11 temperature and humidity sensor:Temperature measurement range: 0 to 50 degrees Celsius with an accuracy of +/−2 degrees CelsiusHumidity measurement range: 20% to 90% RH (Relative Humidity) with an accuracy of +/−5%Operating voltage: 3.3 V to 5 V DCCurrent consumption: 2.5 mA max during conversion, 0.5 mA standbySignal transmission distance: Up to 20 mSampling rate: 1 Hz (1 reading per second)Response time: less than 5 sDimensions: 23 mm × 12 mm × 5 mm (L × W × H)Digital output: Data is transmitted using a single-wire serial interface with a timing protocol that includes a start signal, data bits, and a checksum for error checking.

The DHT11 sensor is a low-cost, basic temperature and humidity sensor that is suitable for many simple projects. However, if you require higher accuracy, faster response time, or other features, you may need to consider a more advanced sensor.”

The bot was able to summarize the applicability of the sensor in an efficient manner, but a mistake was found. The response time was not accurate. The datasheet gives 6 s minimum, 10 s typical and 15 s max for this parameter [26], compared to the bot’s version of “less than 5 s”. It was suggested that DHT11 has different datasheets available on the web, but an alternative, widely available datasheet version was also backing up our finding with precise response time values [27]. We asked the same question for DS18B20 sensor too Appendix A, where we found that the maximum of the current draw was imprecise for the suggestions (1 mA suggested value vs. 1.5 mA in real life). For two other sensors (TLS2561, MQ-135), the operating voltages, active power consumption and operating temperatures were imprecise (for full details, see Appendix A), in the meanwhile, staying in similar range as the datasheet values. For TLS2561, ChatGPT gave an “Operating voltage range of: 2.7 V to 3.6V DC”, which is actually 0 V to 3.8 V according to datasheet [28]. It was found that the “2.7 V to 3.6 V DC” was received from an application note [29], where the “voltage range” noting is similarly confusing as it is missing the lower boundary of the range. For MQ.135: ChatGPT gave “Operating temperature: −10 to 50 °C”, which is actually max. 45 °C [30].

Overall, this information can be crucial during design, so while ChatGPT can focus its interest on specific components, many details need validation by human hand. It is suggested that the bot was trained on older data or misleading information was found in the training dataset. Altogether it can be concluded that most of the given data were usable though some responses contained imprecise information in similar, easily confusing or deceiving range as original values found in technical specifications.

### 3.3. Workflow and Code

To aid embedded coding, ChatGPT was asked how to connect a DS18B20 temperature sensor to the Arduino [31]. The component was chosen randomly from the previous set of results discussed in Section 3.2. The answer was the following.

***ChatGPT****:* “To connect a DS18B20 temperature sensor to an Arduino, you will need the following components:DS18B20 temperature sensor4.7 k ohm resistorBreadboardJumper wires

Here are the steps to connect the DS18B20 temperature sensor to the Arduino:Connect the Vcc pin of the DS18B20 sensor to the 5 V pin on the Arduino.Connect the GND pin of the DS18B20 sensor to the GND pin on the Arduino.Connect the data pin (middle pin) of the DS18B20 sensor to a digital input/output (I/O) pin on the Arduino. In this example, we will use digital pin 2.Connect a 4.7 k ohm resistor between the data pin and the 5 V pin on the Arduino.Connect the Arduino to your computer via USB cable.Open the Arduino IDE and create a new sketch.In the sketch, include the OneWire and DallasTemperature libraries. These libraries can be installed from the Arduino IDE Library Manager.Initialize the OneWire and DallasTemperature libraries in the setup() function of the sketch.In the loop() function of the sketch, read the temperature data from the DS18B20 sensor using the DallasTemperature library functions. The code would look something like this:

In this code, we first define the data pin (digital pin 2) as ONE_WIRE_BUS. Then we initialize the OneWire and DallasTemperature libraries in the setup() function. In the loop() function, we first use the requestTemperatures() function to request the temperature data from the DS18B20 sensor. Then we use the getTempCByIndex(0) function to get the temperature in degrees Celsius. Finally, we print the temperature to the serial monitor and add a delay of 1 s before the next reading.

10.Upload the sketch to the Arduino board, and open the serial monitor to see the temperature readings.”

According to the given response, we validated the steps by the datasheet. The component list was valid, as well the connections of the pins and components. Even highlighting the 1-Wire communication was correct. The official datasheet backs up our findings about the response [31]. The suggested code is also working, but minimal knowledge of the IDE (Integrated Development Environment) is required to assess the steps given by the bot. We also did the query with the DHT11 Appendix A, and we received again, convincing points. The datasheet was suggesting that the pin alignment was correct [32]. However, there was a DHT library recommendation, which all authors agreed upon, that was not straightforward to understand and include with the work. Overall, it was found that more preliminary knowledge is required to use the second query, and it was found that the depth of the solution might vary from use case to use case.

To the question about the origins of the suggestions, ChatGPT responded with a sound answer:

***ChatGPT****:* “I wrote the code example based on my knowledge and experience with using the OneWire and DallasTemperature libraries to read temperature data from a DS18B20 sensor. However, the code structure and some of the function calls may be similar to code examples found in the official Arduino documentation, online forums or tutorials.”

The generated code can be found in ***CodeForDS18B20.cpp*** in the Appendix A dataset of the article.

The question regarding the use of an MQ-4 gas sensor was also asked from the bot. This time the results were considerably less reliable. ChatGPT cited the datasheet of the MQ-4 sensor for a formula [33], where the datasheet did not contain such information. So, human validation was again required at this point to validate the data provided by the bot. From this aspect, the formula was not included in the official support materials, so we can claim this part unreliable.

In the codes, commenting was also found for better understanding; however, the comments were not consequential in their amount and detail. It is a big advantage however that the tool was aware of commenting.

Altogether, ChatGPT can give partially plausible solutions for simple hardware and software tasks, like interfacing a controller and a sensor, but it also gave false findings, which elevates the need of human control over the results.

### 3.4. Literature Survey on the Topic

About the literature survey, the following question was given to ChatGPT: “Could you do a literature review on smart home control with Arduino for a research paper?” ChatGPT responded with a general introduction on the topic of Arduino in relation with smart home control and gave three literature references as examples. For example:

***ChatGPT****:* “One study by Varghese et al. (2017) presented an Arduino-based smart home system that used a variety of sensors to monitor the temperature, humidity, gas levels and motion in a house. The system also included automatic control of lights and fans based on occupancy and environmental conditions. The authors reported that the system was able to improve energy efficiency and reduce utility costs.”

When we asked about the citations, ChatGPT provided a plausible response at first glance. The following citation was presented for the article above.

***ChatGPT****:* “Varghese, J., Jayachandran, S., & Jayachandran, S. (2017). An Arduino based smart home system. International Journal of Pure and Applied Mathematics, 114(10), 505–514.”

Looking for the given citation in widely used search-engines (Google Scholar, Academia.edu) and in the article list of the given journal, it was found that this paper did not ever exist. The names of the authors were found in similar research topics, but it was an initial red flag that the second and third authors were the same (Jayachandran, S.). Also, the title is a fabricated one, as well as the rest of the bibliographic information. In the next step, ChatGPT was asked to give references to scientific literature (recommend articles) regarding Arduino as a sensor controller (also including the DOI numbers of the papers). The response was a list of five references, with similarly false information. During the investigations, an interesting aspect was revealed about the following suggested article:

***ChatGPT****:* “J. R. de Sousa and M. F. de Sousa, “Data acquisition and control system based on Arduino platform for low-cost greenhouse monitoring,” Measurement, vol. 168, p. 108273, Jan. 2021. DOI: 10.1016/j.measurement.2020.108273”

It turns out, there is no such article available on the web. But the DOI seemed to be in connection with the given journal, which would also fool the unsuspecting eye, believing that the reference is correct. The DOI is an existing one, but it does not refer to the given article. The paper of Ziolowsky et al. has nothing to do with the discussed applied sensors topic [34].

For the question on the actual help for fitting sensors to Arduino, the bot provided five references with short descriptions, and four out of them were actually accessible. One of these examples was not a live link, but the browser forwarded the request to a proper one. The given link “https://www.arduino.cc/en/Tutorial/HomePage (accessed on 29 March 2023)” seemed to forward us to the actual (current) document repository of Arduino [35]. This case needed deeper assessment. The link was fed to the reliable Archive.Org “Wayback Machine”, and it turned out that the archived (March 2023) link is redirected to the actual repository as shown above with the following HTTP response [36]:

“https://www.arduino.cc/en/Tutorial/HomePage |

16:41:50 March 26, 2023

Got an HTTP 302 response at crawl time | Redirecting to…

https://docs.arduino.cc/tutorials/”

According to Archive.org [36], it seemed that 04:07:56, 4 January 2022 was the last timestamp, when the originally suggested link was accessible. This is in line with the communicated capability of ChatGPT 3.5, that is, relying on a 2021 learning database—the bot could not know that the tutorials were removed to an alternative website.

### 3.5. An Outlook and Comparison with Bing

While the basic assumption to compare Bing to the closed database-trained ChatGPT 3.5 is not a balanced act, we asked the questions on controllers, argument on them, sensor types, specific information on getting started and specific literature. All details are available in Appendix A.

We found that Bing gave us five valid controllers (when asked about 4–6 different types, pointing to a middle solution in quantity). The arguments and information were valid, but the descriptions were a bit shorter. Out of five examples, four were the same with the recommendations of GPT. Also, Arduino was the primary choice here as well.

As for sensor types, Bing gave eight different sensors, where most of them were healthcare based ones. So the focus was much more precise in the case of ChatGPT. For a more specific query on temperature sensors, five different types were listed. DS18B20 was highlighted again as a widely used one. Bing could recommend altogether 17 valid links for tutorials and videos. This aspect is clearly realized by the connection to the web.

While we asked for at least five literature references, we were given five articles. Four articles were available of the five ones, but three articles were written by other authors as it was stated by the bot. Also, all of the DOIs were imprecise. Most links were cited from ResearchGate, which might come from the bias to search in open access materials and libraries. It can be stated that even this tool has problems, so literature survey with the tool is not acceptable from academic precision aspects.

## 4. Discussion

The summary of our findings is presented in Figure 2 and in the performance metrics Table 1, based on the work of Lo [37]. We found that ChatGPT was able to narrow down an initial assumption in developing a smart home project with a proper choice of controllers and sensors. The bot gave superficial but altogether usable recommendations for a controller, choosing from a wide range of items, limited by the its training. It also described a wide range of sensors for the project, with relevant descriptions accessible to even a beginner practitioner in the field.

As for the sensors and specifications, the bot was able to sort out different sensor types for the task, which were fitting to the request, and which are still on the market. The sorting of the sensors was correct from the aspect of the price, but the listings (or groupings) were inconsistent from the aspect of sensor type (by measurement principle) and actual sensor component type. We also found that most of the sensor specifics were correct, but around 10% of the given parameters were imprecise (albeit in range, close to the actual values). The bot was able to follow the discussion via initial assumptions fixed in the queries (such as following up with the Arduino platform). The bot was also giving popular examples and was not focusing on very specific components but widely available ones. If the training dataset is new enough, mistakes in given data could be less, and exact parameters could be more precise; however, various versions of datasheets floating around (e.g., in the case of cheap Far East sensors) might also leave space for confusion.

The workflow of interfacing a sensor module to a controller was covered by the ChatGPT in a partially proper manner. It was able to give appropriate design solutions for physically interfacing the sensor module and also generated a code that provided correct syntax and references to the application of the given sensor module. The request was repeated with another sensor component, where the answers were similarly convincing, but contained minor errors. It must be noted that the task was relatively simple. However, at this level, the bot can help engineers in the learning phase or give an immediate response for a task, which would eventually require a significant amount of working hours.

The capability of the bot for a literature survey is found to be unreliable. It recommends seemingly correct and relevant citations for given technical themes. But all of them are fake, fabricated by relevant keywords, existing authors, journal names and DOIs, which are real but incorrect for the actual examples. It is suggested that in the nature of this LLM, the bot is assembling keyword-based information and cannot handle large, coherent data, like a literature citation. Or simply, the literature is not available to the bot due to commercial nature. In a very recent study, there is a suggestion, that the ChatGPT is more like a creative tool in case of literature analysis than exact documentation assistant [38]. However, it further raises questions about how the information is handled in other scientific fields; as the aforementioned review material mentions [39], most papers seem optimistic from the data organizing side capabilities (e.g., in hypothesis validation or the identification of gaps and inconsistencies in current knowledge). The fabricated literature responses had an added value though: It contained coherent listings of thematically correct key-words, which could further aid the literature survey of a human learner or a beginner in the field. It is important to note that for more technical or commercial related information (such as datasheets and tutorials), ChatGPT could provide four accessible links out of five. One link was an older one, but it was automatically redirected to a working page by the host.

It has to be also noted that the flow of the discussion cannot be reinstated or repeated in the same manner, so further information and branching discussions are given to the user distinctively from time to time. Our experiment presented a use case that is not classically reproducible in the traditional manner. This also makes it more difficult to assess the application of the tool from both technical and societal aspects.

Overall, our results are clarifying the technical depth and precision of the literature survey shown in [5,6,11], showing that exact literature survey, or “searching” can be unreliable. The partially acceptable results with minor failures were similar to be found in [4]. As we found, the accessibility and applicability of the tool points to a curious recommendation, which is similar to [18,19,20]. The performance analysis was similar to other ChatGPT use cases [37].

## 5. Conclusions

In this article, we investigated the applicability of the novel LLM-based ChatGPT for electrical engineering development of a smart home project. ChatGPT followed our queries with a consistent manner, focusing on the controller and narrowing down the question to different sensor types, their interfacing, some exemplary software and related literature.

It was found that the sensor recommendations were valid, but part of the given specification, prices, hardware-software integration and the resulting code were slightly imprecise, which needed human validation. Furthermore, the listings were inconsistent from the sensor type and commercial module type aspects.

It was found that, by requesting actual scientific literature (articles, journal papers), the results were fabricated, fake and unreliable. However, with limited usability, technical documentation was referenced correctly by ChatGPT, and the key words were found to be accurate.

Interestingly, it was found that Bing (connected to the web) made similar performance as the older ChatGPT 3.5 version, where controller choice was acceptable, sensor technical details were containing minor problems, and literature survey was unacceptable.

For the future, similar tasks should be investigated with the continuously evolving versions and the competitors, such as the recently launched Bing or the Bard platform. The work could be continued with the inclusion of PCB (printed circuit board) design and related simulations. Our work does not tackle the ethical side and fair, responsible use, which is a societal aspect. Concerns are high regarding the application of such tools; even leading professionals are asking for a “pause time” from AI developers to catch up with regulations. Also, it is suggested to focus on information technology and engineering R&D aspects during the development of further releases. As the tool is coming from the same broadly interpreted profession, as our field, it would be important for the tool to meet requirements in data precision (especially with literature and technical information), so that it does not include errors to the developers working with it.

The takeaway message is that, with proper regulations and future improvements, chatbots can be an effective device in electronics development projects, as a supplementary tool in the available, traditional toolset.

## Figures and Tables

**Figure 1 sensors-23-04879-f001:**
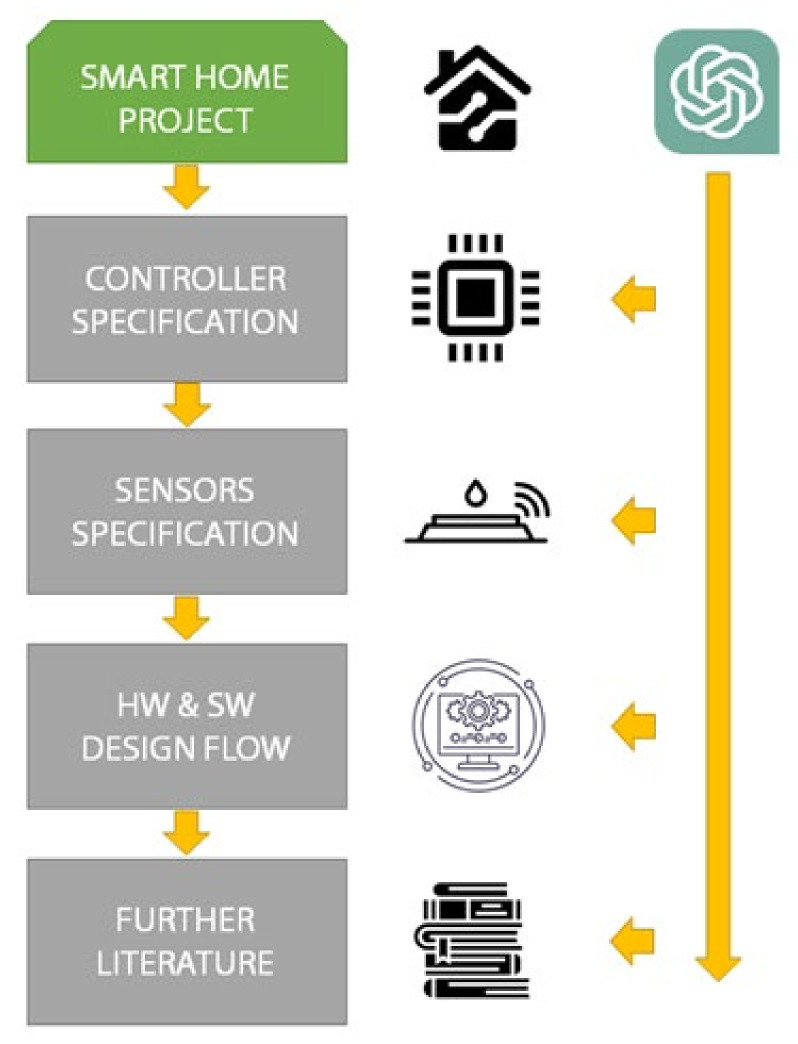
Workflow with ChatGPT in the use case of building a smart home device with sensors.

**Figure 2 sensors-23-04879-f002:**
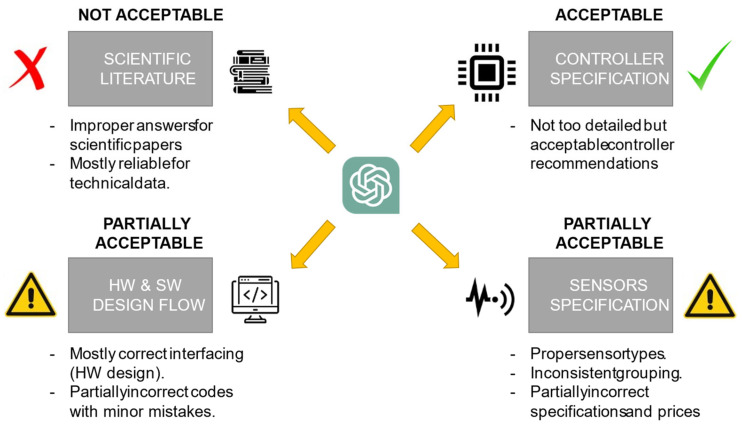
Summarizing our findings regarding main aspects and the acceptability of the given answers by ChatGPT.

**Table 1 sensors-23-04879-t001:** Performance evaluation of ChatGPT-3.5 across different subjects.

Subjects	Overall Performance	Comments
Choice of controller for smart home application	**Acceptable**	Outstanding choice with proper details on different types, Arduino was the leading choice, understandable discussion, usable but shallowly detailed information.
Sensor types	Partially acceptable	Type choice was proper, with inconsistencies in listing and presentation. No commercialization was apparent.
Sensor specification	Partially acceptable with minor technical errors	Specifications were mostly ok, minor problems were found in specification parameters (only one per type) and pricing. Probably due to limited availability to the internet.
Hardware interfacing	Partially acceptable	Interfacing details were almost acceptable with minor inconsistencies in presentation. The descriptions were written with varied required basic knowledge for understanding.
Code	Partially acceptable with occasional technical errors	Correct syntax, uneven commenting, minor addressing issues.
Literature	**Unacceptable**	Totally unacceptable for academic demands. Fabricated titles with real authors and fake, sometimes working, but not connected DOIs. Technical data is addressed in a more reliable form.

## Data Availability

Appendix A is available at: http://real.mtak.hu/id/eprint/163609 (accessed on 5 April 2023).

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
