# Peer review of "Can ChatGPT Help in Electronics Research and Development? A Case Study with Applied Sensors"

_sensors, 2023, doi:10.3390/s23104879_

Round 1
Reviewer 1 Report
I appreciate the effort and knowledge the author put into the research. To improve the quality of the manuscript, I recommend the author improve the following things.
ChatGPT is a language bot, I expect the author to compare it with other language models. Line 174-175 is unacceptable.
Create a performance metric to evaluate the performance of the case study.
Figure 1: Should be revised (The icons are not related to the actual representation).
Include related works section separately on languages learning and AI modeling.
Include contribution of the research in the introduction section.
Don't include citations in the conclusion section.
I recommend the author to change the question with the same meaning, and list out the results.
When designs and sensor details are partially accepted, authors should explain which is wrong and why. This manuscript cannot be accepted without proper comparisons.
Minor editing of English language is required.
Author Response
Dear Reviewer, your comments were noted, and we are thankful for your insight on the topic.
As we found, there might be a confusion with the availability of supplementary materials, which would have helped the comments on actual comments -> the system of our academy’s repository put the uploaded material into „review phase”, so it might became unavailable during the time of the „Sensors” journal review process. Now we give a temporary link as well to [SupMat] which would help more in-depth understanding.
The corrections are noted with blue in the paper.
R1C1: I appreciate the effort and knowledge the author put into the research. To improve the quality of the manuscript, I recommend the author improve the following things.
R1A1: Thank you for the positive feedback, it reassures us that we are on the right track.
R1C2: ChatGPT is a language bot, I expect the author to compare it with other language models. Line 174-175 is unacceptable.
R1A2: Thank you for highlighting this, and we agree partially. In the moment of the writing, 3.5 version was available, and V4 was not out in the field. During the review round, V4 was introduced but behind a paywall. In the meantime the same happened to Bard and Bing as well. Also we could not reach Bard without a VPN as it was not officially supported by Google (meaning that worldwide access was still scarce.). However Bing was available, so we expanded with a short comparison chapter at the end, also rewriting our stance on what you found unacceptable. We hope it is acceptable now in this form and with these arguments.
Also please note that we did not want to bloat the paper with additional quotations from the queries and answers, but all info is available in [SupMat].
„While the next version of ChatGPT (GTP-4) is supposed to have better performance, it just became available during our investigations and was locked behind a paywall. Furthermore, controversies were noted, and concerns were raised regarding the new version of the tool (with an eventual ban of it in Italy). [23] So, we aimed to stick to the more well-known version of the tool. Also, during the manuscript composition, Bard (Google) was not available in the country, meaning that general accessibility is still an issue. Bing (Microsoft) bots became available with interfacing to the internet [24], and it was freely available, so we decided to add a short (but not comprehensive) comparison between 3.5 and Bing.”
„3.5. An outlook and comparison with Bing
While the basic assumption to compare Bing to the closed database-trained ChatGPT 3.5 is not a balanced act, we asked the questions on controllers, argument on them, sensor types, specific information on getting started and specific literature. All details are available in [SupMat].
We found that Bing gave us five valid controllers (when asked about 4-6 different types, pointing to a middle solution in quantity). The arguments and information was valid, but the descriptions were a bit shorter. Out of five examples, four were the same with the recommendations of GPT. Also, Arduino was the primary choice here as well.
As for sensor types, Bing gave eight different sensors, where most of them were healthcare based ones. So the focus was much more precise in the case of ChatGPT. For a more specific query on temperature sensors, five different types were listed. DS18B20 was highlighted again as a widely used one. Bing could recommend altogether 17 valid links for tutorials and videos. This aspect is clearly realized by the connection to the web.
While we asked for at least five literature references, we were given five articles. Four articles were available of the five ones, but three articles were written by other au-thors as it was stated by the bot. Also, all of the DOIs were imprecise. Most links were cited from ResearchGate, which might come from the bias to search in open access mate-rials and libraries. It can be stated that even this tool has problems, so literature survey with the tool is not acceptable from academic precision aspects.”
R1C3: Create a performance metric to evaluate the performance of the case study.
R1A3: We looked for a proper performance metric to evaluate our work, and we found a relevant solution in [https://doi.org/10.3390/educsci13040410] in a similar paper dealing with ChatGPT, and which was published in MDPI. Please find the detailed summary in Chapter 4, Table1.
R1C4: Figure 1: Should be revised (The icons are not related to the actual representation).
R1A4: We have changed the sensor and the design flow icons. While all were searched and found as actual representation of the given titles, we felt that these two might really be confusing for many. So you can find Figure 1 as a revised format in the paper now. This was also carried on to Fig. 2.
R1C5: Include related works section separately on languages learning and AI modeling.
R1A5: The introduction is reorganized in a more specific manner to separate different related works. Please find main changes in blue in the given chapter.
R1C6: Include contribution of the research in the introduction section.
R1A6: The introduction is expanded upon the last paragraph, to further show our contribution tot he field.
R1C7: Don't include citations in the conclusion section.
R1A7: Good point, we omitted the citation from the conclusion.
R1C8: I recommend the author to change the question with the same meaning, and list out the results.
R1A8: Thank you to highlight this. As we see, this point is aiming to show the repeatability of the GPT tool. As it was noted, GPT could give slightly different answers for the same query at times, so we think that applying this kind of repeatability check is not applicable for the current case.
R1C9: When designs and sensor details are partially accepted, authors should explain which is wrong and why. This manuscript cannot be accepted without proper comparisons
R1A9: This point is further detailed by the recommended addition of performance metrics. Also please find inclusions with blue to Chapter 3 which addresses these issues, where we tried to aim for a more extensive discussion.
There might be a misunderstanding on this point as well, as the discussions can be followed with the given supplementary materials in mind, in much deeper details. We also emphasize connection to this, where it is neccessary.
„which is wrong and why.” -> We further emphasized these parts. like shown below:
„For two other sensors (TLS2561, MQ-135), the operating voltages, active power consumption, and operating temperatures were imprecise (for full details, see [SupMat]), in the meanwhile staying in similar range as the datasheet values. For TLS2561 ChatGPT gave an “Operating voltage range of: 2.7V to 3.6V DC” which is actually 0 V to 3,8V according to datasheet [29]. It was found that the “2,7V to 3,6V DC” was received from an application note [30], where the “voltage range” noting is similarly confusing, as it is missing the lower boundary of the range. For MQ.135: ChatGPT gave “Operating temperature: -10 to 50°C “ which is actually max. 45°C [31].”
Thank you for your review again!

Reviewer 2 Report
ChatGPT is very popular at the moment, and many people are very fond of it. The authors have carried out a study on the application of ChatGPT and pointed out its advantages and disadvantages from a fair point of view, which is very worth advocating. However, the paper is only a validation of the ChatGPT application, and lacks innovation and the authors’ own insights and opinions. My comments are as follows:
1. To narrow down the focus, the authors gave two controllers as an initial choice. For the well-established topic, this is a bit easy for bot. I think it will be more convincing if the initial choice is three or four.
2. The specifications or other more detailed information given by ChatGPT about the sensor of interest is very relevant to the training dataset. Does it make sense to talk about the accuracy of specifications in this case? If the training dataset is cutting-edge or new enough, the specifications given will be exactly correct, right?
3. ChatGPT was asked how to connect a DS18B20 temperature sensor to the Arduino in 3.3 Workflow and code. The relevant specifications of DHT11 were given by ChatGPT in 3.2 Sensors and specifications. The authors also analyzed their accuracy. What about DS18B20? Why did you not analyze the specifications accuracy of DS18B20 first and then gave the connection scheme by ChatGPT? It seems more logical.
4. ChatGPT was asked to do a literature review on smart home control with Arduino, and unfortunately the citation given by ChatGPT is incorrect. It would be more innovative if you could add the reasons why this happens.
5. The author only analyzed the accuracy of one sensor specification, and draw a conclusion about the accuracy of ChatGPT in Discussion or Conclusions are not rigorous. Other parts have similar problems.
6. It is suggested that the authors could put forward some improvement suggestions on ChatGPT on the basis of application verification, which will also increase the innovation of the paper.
7. References are marked in different positions in the text, some in the middle of the sentence and some at the end of the sentence, which gives reader an uncomfortable feeling. It would be even better if the locations are uniform.
8. I think the academic value of this manuscript is enough for readers.
The quality of English language of this manuscript is fine.
Author Response
Dear Reviewer, your comments were noted, and we are thankful for your insight on the topic.
As we found, there might be a confusion with the supplementary materials, which would have helped better understanding-> the system of our academy’s repository put the uploaded material into „review phase”, so it might became unavailable during the time of the „Sensors” journal review process, so we give a temporary link as well to [SupMat] which would help more in-depth understanding.
The related corrections are noted with purple in the paper.
R2general1: The quality of English language of this manuscript is fine.
R2Ag1: Thank you for noting.
R2general2: ChatGPT is very popular at the moment, and many people are very fond of it. The authors have carried out a study on the application of ChatGPT and pointed out its advantages and disadvantages from a fair point of view, which is very worth advocating.
R2Ag2: Thank you for noting. It emphasizes that we are on the right track.
However, the paper is only a validation of the ChatGPT application, and lacks innovation and the authors’ own insights and opinions. My comments are as follows:
With the three reviews and the corrections hopefully most of your concerns are addressed.
R2comment1. To narrow down the focus, the authors gave two controllers as an initial choice. For the well-established topic, this is a bit easy for bot. I think it will be more convincing if the initial choice is three or four.
R2A1: Thank you for noting. In the revision we asked the bot to give 4-6 different versions. It gave us 6 (aimed for the higher number), and the discussion on this topic was extended in each relevant chapters. All in all, the bot gave convincing and good information from this aspect of the work, still after 6 recommendations.
The larger addition can be seen in 3.1 (in SupMat as well) and you can see further smaller comments in purple later connected to this topic.
R2C2: The specifications or other more detailed information given by ChatGPT about the sensor of interest is very relevant to the training dataset. Does it make sense to talk about the accuracy of specifications in this case? If the training dataset is cutting-edge or new enough, the specifications given will be exactly correct, right?
R2A2: This is true, however we wanted to test the snapshot which was available during the time of the experiments and the manuscript writing.
We further help understanding according to your comment, while it raised a further idea in our heads.
„If the training dataset is new enough, mistakes in given data could be less, and exact parameters could be more precise, however various versions of datasheets floating around (e.g. in the case of cheap Far East sensors) might also leave space for confusions.”
R2C3: ChatGPT was asked how to connect a DS18B20 temperature sensor to the Arduino in 3.3 Workflow and code. The relevant specifications of DHT11 were given by ChatGPT in 3.2 Sensors and specifications. The authors also analyzed their accuracy. What about DS18B20? Why did you not analyze the specifications accuracy of DS18B20 first and then gave the connection scheme by ChatGPT? It seems more logical.
R2A3: Good point, we tried to specify both DS and DHT sensors from both sides, and we came to similar conclusions – albeit, the discussion is full and logical now, as the reviewer suggested.
„We asked the same question for DS18B20 sensor too [SupMat], where we found that the maximum of the current draw was imprecise for the suggestions (1 mA suggested value vs 1.5 mA in real life).”
„We also did the query with the DHT11 [SupMat], and we received again, convincing points. The datasheet was suggesting that the pin alignment was correct. [33] However, there was a DHT library recommendation, which all authors agreed upon, was not straightforward to understand and include to the work. Overall, it was found that more preliminary knowledge is required to use the second query, and it was found that the depth of the solution might vary from use case to use case.”
R2C4: ChatGPT was asked to do a literature review on smart home control with Arduino, and unfortunately the citation given by ChatGPT is incorrect. It would be more innovative if you could add the reasons why this happens.
R2A4: „The capability of the bot for a literature survey is found to be unreliable. It recommends seemingly correct and relevant citations for given technical themes. But all of them are fake, fabricated by relevant keywords, existing authors, journal names, and DOIs, which are real but incorrect for the actual examples. It is suggested that the in the nature of this LLM, the bot is assembling keyword-based information, and can not handle large, coherent data, like a literature citation. Or simply, the literature is not available to the bot due to commercial nature. In a very recent study, there is a suggestion, that the ChatGPT is more like a creative tool in case of literature analysis, than exact documentation assisatnt [38]. However, it further raises questions, how the information is handled in other scienific fields; as the aforementioned review material mentions [38], most papers seem optimistic from the data organizing side capabilities (e.g. in hypothesis validation or the identiication of gaps and inconsistencies in current knowledge). The fabricated literature responses had an added value though: it contained coherent listings of thematically correct key-words, which could further aid the literature survey of a human learner or a beginner in the field. It is important to note that for more technical or commercial related information (such as datasheets and tutorials), ChatGPT could provide four accessible links out of five. One link was an older one, but it was automatically redirected to a working page by the host. .”
R2C5: The author only analyzed the accuracy of one sensor specification, and draw a conclusion about the accuracy of ChatGPT in Discussion or Conclusions are not rigorous. Other parts have similar problems.
R2A5: Please note that in the [SupMat] there is more information to be found. This is further emphasized now in the paper, as Reviewer 1 suggested as well (in blue color). Also R2C3 and the answer hopefully further helps this issue, and deepens the discussion.
Also we added a more detailed evaluation in Discussions, hopefully also improving the merit of the paper.
R2C6: It is suggested that the authors could put forward some improvement suggestions on ChatGPT on the basis of application verification, which will also increase the innovation of the paper.
R2A6: Thank you for noting. It is a very good point. We added this to our work.
„Also, it is suggested to focus on information technology and engineering R&D aspects during the development of further releases. As the tool is coming from the same broadly interpreted profession, as our field, it would be important for the tool to meet requirements in data precision (especially with literature and technical information), so that it does not include errors to the developers working with it.”
R2C7: References are marked in different positions in the text, some in the middle of the sentence and some at the end of the sentence, which gives reader an uncomfortable feeling. It would be even better if the locations are uniform.
R2A7: Thank you, it is an inconsistency – we uniformed all citations. They are now appearing only before complex sentence commas (where we wanted to focus the reference to a closer point) and after punctuations in more general parts.
Find purple changes around the paper at reference markings.
R2C8: I think the academic value of this manuscript is enough for readers.
R2A8: Thank you for noting. It is a very important comment in the manner of these review rounds.

Reviewer 3 Report
The authors investigated the applicability of ChatGPT AI in electronics research and development via a case study of applied sensors in embedded systems. The initial electronics-development tasks of a smart home project were prompted to the ChatGPT system to find out its capabilities and limitations. We wanted to get detailed information on the central processing controller units and actual sensors usable for the specific project, their specifications, and recommendations on the hardware and software design flow additionally. Furthermore, an extensive literature survey was requested to see if the bot could offer scientific papers covering the given topic. It was found, the ChatGPT responded with proper recommendations on controllers. However, the suggested sensor units, the hardware and software design were only partially acceptable, with occasional errors in specifications and generated code. The results of the literature survey showed that non-acceptable, fabricated citations (fake authors list, title, journal details and DOI – Digital Object identifier) were presented by the bot. The paper provides a detail and qualitative analysis of the forementioned aspects with providing the query set, the generated answers, and codes as supplied data. . I have the following comments
1- Please underscore the scientific value added of your paper in your abstract and introduction.
2- The contribution is not clear in the introduction part. Introduction should be clearly stated research questions and targets first. Then answer several questions: Why is the topic important (or why do you study on it)? What are research questions? What has been studied? What are your contributions? Why is to propose this particular method (This must come from Literature discussion)? .
3- The paper should describes separately the problem statement, the motivation, and the real contribution.
4- The authors should strengthen their argument with regards to the scientific novelty and also justify the need to investigate the applicability of ChatGPT AI in electronics research and development.
5- The related work section is not well written; the authors should insert the results of the previous related works and make a critical analysis by introducing the weaknesses or shortcomings of these works.
6- The results are not enough to show the efficiency of the proposed work, so, please extend the results parts to include a further experiments with a comparison with other works.
Author Response
Dear Reviewer, your comments were noted, and we are thankful for your insight on the topic.
As we found, there might be a confusion with the supplementary materials, which would have helped the comments on actual comments -> the system of our academy’s repository put the uploaded material into „review phase”, so it might became unavailable during the time of the „Sensors” journal process, so we give a temporary link as well to [SupMat] which would help more in-depth understanding.
The related corrections are noted with yellow in the paper.
R3Cgeneral The authors investigated the applicability of ChatGPT AI in electronics research and development via a case study of applied sensors in embedded systems. The initial electronics-development tasks of a smart home project were prompted to the ChatGPT system to find out its capabilities and limitations. We wanted to get detailed information on the central processing controller units and actual sensors usable for the specific project, their specifications, and recommendations on the hardware and software design flow additionally. Furthermore, an extensive literature survey was requested to see if the bot could offer scientific papers covering the given topic. It was found, the ChatGPT responded with proper recommendations on controllers. However, the suggested sensor units, the hardware and software design were only partially acceptable, with occasional errors in specifications and generated code. The results of the literature survey showed that non-acceptable, fabricated citations (fake authors list, title, journal details and DOI – Digital Object identifier) were presented by the bot. The paper provides a detail and qualitative analysis of the forementioned aspects with providing the query set, the generated answers, and codes as supplied data.
R3Ageneral: Thank you for the proper summary, it helps us that the reviewer was able to summarize the most improtant points, so that the paper is followable in its communication and messages.
R3C1: Please underscore the scientific value added of your paper in your abstract and introduction.
R3A1: Please find the improved abstract below.
Abstract: „In this paper, we investigated the applicability of ChatGPT 3.5 AI in electronics research and development via a case study of applied sensors in embedded electronic systems, which topic is rarely mentioned in the recent literature, thus providing new insight for professionals and academics. The initial electronics-development tasks of a smart home project were prompted to the ChatGPT system to find out its capabilities and limitations. We wanted to obtain detailed information on the central processing controller units and actual sensors usable for the specific project, their specifications, recommendations on the hardware and software design flow. Furthermore, an extensive literature survey was requested to see if the bot could offer scientific papers covering the given topic. It was found, the ChatGPT responded with proper recommendations on controllers. However, the suggested sensor units, the hardware and software design were only partially acceptable, with occasional errors in specifications and generated code. The results of the literature survey showed that non-acceptable, fabricated citations (fake authors list, title, journal details and DOI – Digital Object identifier) were presented by the bot. The paper provides a detailed qualitative analysis, a performance analysis and critical discussion of the aforementioned aspects with providing the query set, the generated answers, and codes as supplied data with a performance analysis The goal is to provide added value to electronics researchers and developers, if trying to reach out for the tools in their profession.”
R3C2:- The contribution is not clear in the introduction part. Introduction should be clearly stated research questions and targets first. Then answer several questions: Why is the topic important (or why do you study on it)? What are research questions? What has been studied? What are your contributions? Why is to propose this particular method (This must come from Literature discussion)? .
R3A2: Thank you for the comment, this was addressed with an additional paragraph along your and another reviewer’s suggestions:
„To see, how efficiently can ChatGPT help in electronics research and development, we adapted the methodologies and approaches presented in the previous papers and investigated the applicability of ChatGPT for electrical engineering, design and development, where our focus was positioned to a widely studied topic: smart home applications with sensors. The question was: can ChatGPT be a tool or a useful companion for an electrical engineer developer? The paper presents the experimental methodology, the query runs with the bot with the supplemented data, and the discussion of the results. The main novelty of our work is to show the applicability of the tool in an area which was not discussed before, and our findings might help discussions and arguments regarding the application from the professional industrial level to education and academic sciences”
R3C3:- The paper should describes separately the problem statement, the motivation, and the real contribution.
R3A3: Thank you for the comment, please find the revised introduction and other corrections made for the request of R1 and R2 reviewers.
R3C4:- The authors should strengthen their argument with regards to the scientific novelty and also justify the need to investigate the applicability of ChatGPT AI in electronics research and development.
R3A4: Thank you for the comment -> please see the comments to R1 and R2 reviewers. Most of our efforts were spent to show an analysis of the results, to further deepen the criticism by pointing out exact imprecisions done by ChatGPT, to include new, more in-depth queries and so on. We added further ideas for discussion part as well. We hope that the arguments are stronger now.
R3C5:- The related work section is not well written; the authors should insert the results of the previous related works and make a critical analysis by introducing the weaknesses or shortcomings of these works.
R3A5: Actually the related works are reorganized to show a better overwiew. Also the results are criticised now. Please find blue and for more criticism, new yellow color changes in Chapter 1.
R3C6:- The results are not enough to show the efficiency of the proposed work, so, please extend the results parts to include a further experiments with a comparison with other works.
R3A6: Thank you for the comment -> please see the comments to R1 and R2 reviewers as noted before, also the extensions, and also our new experments with the inclusion of Bing. Also please see the final paragraph of discussion:
„Overall, our results are clarifying the technical depth and precision of the literature survey shown in [5], [6] and [14], showing that exact literature survey, or "searching" can be unreliable. The partially acceptable results with minor failures were similar to be found in. [4] As we found, the accessibility and applicability of the tool points to a curious recommendation, which is similar to [19], [20] and [21]. The performance analysis was similar to other ChatGPT use cases. [38]”

Round 2
Reviewer 1 Report
Revised version looks good now.
Reviewer 2 Report
The authors have try their best to improve the manuscript according to the reviewer's comments.
Quality of English language is acceptable.
Reviewer 3 Report
No comments.